# Evaluation of the Tibial Plateau–Patella Angle (TPPA) in Dogs

**DOI:** 10.3390/ani14121798

**Published:** 2024-06-16

**Authors:** Nedim Zaimovic, Dragan Lorinson, Karin Lorinson, Alexander Tichy, Barbara Bockstahler

**Affiliations:** 1Small Animal Surgery, Department for Companion Animals and Horses, University of Veterinary Medicine, 1210 Vienna, Austria; 2Chirurgisches Zentrum für Kleintiere Dr. Lorinson, 2331 Vösendorf, Austria; ordination@vet-lorinson.com (D.L.); karinunddragan@hotmail.com (K.L.); 3Platform Bioinformatics and Biostatistics, Department for Biomedical Services, University of Veterinary Medicine, 1120 Vienna, Austria; alexander.tichy@vetmeduni.ac.at; 4Section of Physical Therapy, Small Animal Surgery, Department for Companion Animals and Horses, University of Veterinary Medicine, 1210 Vienna, Austria; barbara.bockstahler@vetmeduni.ac.at

**Keywords:** TPPA, proximodistal patellar position, radiography, joint angulation, dog

## Abstract

**Simple Summary:**

The proximodistal patellar position in canine pathologies, including patella luxation, cruciate ligament rupture, and osteoarthrosis, is not fully understood, while this topic is much better understood in human medicine. Various methods are used to confirm the proximodistal patellar position, but they all require calculations and are infrequently used in everyday practice. In a two-step study with three observers, we investigated, for the first time in veterinary medicine, the applicability of a new, simple method derived from human medicine called the tibial plateau–patella angle (TPPA). This method is independent of the knee angle and magnification, does not require any calculations, and consists of just two lines. This study started with cadavers at different stifle angles. Subsequently, we used 100 X-rays at the optimal stifle angle based on these findings. In both study phases, our results revealed strong agreement among observers. The TPPA varied with the stifle angle but remained consistent across the different weight groups. We concluded that the TPPA is directly dependent on the stifle angulation and tends to be lower with a higher stifle extension angle. It could be difficult for some observers to establish the exact caudal border of the tibial plateau while measuring the TPPA. While the TPPA method shows promise, further evaluation, including breed-specific and pathological considerations, is necessary.

**Abstract:**

Estimating a dog’s patellar position involves various methods, which categorize it as norma, alta (high), or baja (low). However, they require various calculations. We aimed to evaluate the clinical applicability of a new method, the tibial plateau–patella angle (TPPA). This could aid in planning patella luxation surgery, estimating the patella position after TPLO and various osteotomies. We conducted a two-step study: first, on 15 stifles without pathologies from nine canine cadavers, and second, using 100 patient X-rays from the archive. Three stifle angle positions (45 ± 5°, 90 ± 5°, and 135 ± 5°) and three weight groups (S, M, and L) were evaluated in the first part of this study. Based on these results, the second part of this study was conducted using 100 pathology-free radiographs at the optimal stifle angle (90 ± 5°) from the archive. All radiographs were measured by three observers with varying levels of experience. Our results indicate that the stifle angle significantly impacted the TPPA, whereby lower values were detected with higher stifle angles, which remained consistent within the weight groups. High inter- and intra-observer agreement was achieved. The physiological TPPA values ranged from 26.7° to 48.8°, remaining consistent within the various weight groups. Observer 3 in Group S exhibited a 20% (insignificant) deviation, possibly due to challenges in determining the caudal point of the tibial plateau. In contrast with humans, TPPA values in dogs are negatively correlated with stifle angles, independent of weight. Our reliable and reproducible protocol suggests the potential benefits of training on small-breed dogs stifles.

## 1. Introduction

The patella is the largest sesamoid bone, sliding in the femoral trochlea, and is crucial to the biomechanics of the stifle joint. Patellar luxation is one of the most common stifle joint pathologies in dogs. Medial patellar luxation is more frequent and classified as congenital, developmental, or traumatic. The underlying issue is not precisely understood; however, most patients show structural abnormalities [1,2,3]. Cranial cruciate ligament disease and patellar luxation, the most common clinical stifle conditions in dogs, lead to the development of osteoarthrosis [4,5]. The influence of the patellar height on clinical pathologies in canines is not precisely defined.

None of the most commonly used indices are ideal for measuring the proximodistal position of the patella [6]. In human medicine, the impact of the patellar height is much more thoroughly researched. For instance, it is known that patella alta can be responsible for stifle pain, patellofemoral instability, a reduced range of motion, and reduced long-term implant survival (total stifle arthroplasty) [7,8,9,10,11,12]. Patella alta is also associated with Osgood–Schlatter disease, a painful patellofemoral syndrome in humans [13,14,15]. In humans, patella baja is a common cause of osteoarthritis (OA) in the patellofemoral joint and represents a significant complication following stifle surgeries, traumas, or immobilization. It can also lead to the shortening of the patellar tendon, fibrosis of the infrapatellar fat pad, intra-articular adhesions, etc. Patella baja is a component of infrapatellar contracture syndrome (IPCS), where the lowering of the patella results from scarring due to inflammation in the peripatellar fat pad, triggered by trauma from surgery [16,17]. Patellofemoral OA is a more common cause of pain in the front of the knee than OA in the tibiofemoral joint [18]. In an experimental study on rats, Bei et al. [19] demonstrated that patella baja may be responsible for the development of osteoarthritic changes.

Radiography is the most common imaging method for orthopedic assessments in small animal practice. Radiological determination of the proximodistal patellar position can be complicated by factors such as osteophytes, joint effusion, an unusual patellar morphology, abnormalities in the tibial tuberosity, and an atypical distal attachment of the patellar tendon [7,20,21,22,23].

In veterinary medicine, several studies have evaluated patellar height in dogs. The most commonly used methods for determining patellar height include the (modified) Insall–Salvati index (ISI) [24,25,26,27], the Blackburne–Peel index (BPI) [6,26], the de Carvalho index (CDI) [6], the ratio of the patellar ligament length (PLL) to the patellar length (PL) (PLL:PL or L:P), and the ratio of the distance from the proximal aspect of the patella (A) to the femur condyle and patellar length (PL) (A:PL) [21,22,28]. 

In 2011, Portner and Pakzad [29] developed the tibial plateau–patella angle (TPPA) for determining patellar height, aiming for it to be independent of human knee angle. The measurement is conducted on a mediolateral X-ray knee image (with a minimum flexion of 30°). The advantages of this method include easy reproducibility, no need for calculations, independence from magnification errors, and the degree of limb flexion [29], making it a simple and fast way to determine the proximodistal patellar position in everyday practice. TPPA measurements in dogs have been performed to compare the pre- and post-operative outcomes in TPLO and TTA patients, but without knowledge of its physiological range [26].

Previous studies in dogs have found that an L:P ratio of >1.97–2.06 or an A:PL ratio of >2.03 indicates patella alta [21,22,28]. Patella alta is highly prevalent in German Shepherd dogs, even without a relevant stifle joint pathology; however, these dogs have twice the risk of developing canine hip dysplasia [25]. Unlike in large dogs, the PLL:PL ratios in small dogs without a stifle joint pathology and with patellar luxation showed no significant difference. According to this study, assessing the proximodistal patellar position using only the PLL:PL ratio may not be feasible [30]. In two different studies, it was concluded that small-breed dogs with medial patellar luxation did not have a proximally positioned patella compared with a control group without medial patellar luxation [30,31].

In Figure 1, we present a simplified scheme of the most commonly used methods for dog stifle measurement, compared to the TPPA.

Since the TPPA in dogs has not been previously described and there is no available evidence for observer reliability, we intended to establish the TPPA as a reliable measurement method for assessing patellar height in dogs. We divided this study into two parts: part one involved a cadaver study, and part two involved a patient study.

In the first part of this study, we aimed to evaluate the impact of the stifle angle on the TPPA, examine observer influence, evaluate the influence of body weight, and determine the optimal stifle angle for measuring the TPPA. The second aim was to evaluate whether observers could reliably identify all the important anatomical points for TPPA measurements.

The main goals of the second part of this study were to measure the TPPA at the optimal stifle angle, establish a physiological range, assess the impact of body weight on TPPA values, and control the identification of anatomical measuring points. We hypothesized that the stifle angulation would influence the TPPA in dogs and both inter- and intra-observer variability would be low. Due to its easy applicability, this method may be useful in everyday practice, such as establishing a normal or pathological patellar height in the pre-operative planning of patellar luxation surgery or the insertion of a stifle endoprosthesis.

## 2. Study Part One: Cadaver Study

### 2.1. Materials and Methods

The first part of this study was a cadaver study that provided TPPA measurements of 15 canine stifle joints derived from 9 dog cadavers. The dogs were euthanatized for reasons unrelated to our study. The cadavers were fresh and free from stifle pathologies. To ensure this, after euthanasia, each cadaver stifle was clinically examined for the persistence of joint effusion, crepitations, cranial/caudal cruciate ligament rupture, patellar luxation, or medial and lateral stifle joint instability. After a stifle was found to be compatible with these criteria, we performed control radiographs (craniocaudal and mediolateral) to rule out malalignments, malformations, or osteoarthrosis. If any of these abnormalities (clinical or radiographic) were persistent, the cadaveric stifle was excluded from this study. We obtained written approval from the owner for every cadaver used.

Based on body weight, three different weight groups were defined (S—small: <15 kg, M—medium: ≥15–<30 kg, and L—large: ≥30 kg). Six of the dog cadavers were females (three spayed), and three were males (two castrated). Four dog cadavers (six stifles) belonged to the S weight group, two (three stifles) to M, and three (six stifles) to L. The mean age of the dog cadavers was 9.1 ± 2.8 (range: 4–13) years.

Mediolateral X-rays at three different angles (45°, 90°, and 135°) were performed on each stifle, with a tolerance level of 5°. The ~45° (40–50°) angulation was defined as Group 1, ~90° (85–95°) as Group 2, and 135° (130–140°) as Group 3. The angulation was directly measured on the cadavers during positioning using a goniometer and later verified in the digital X-ray images using the angle function. All the radiographs provided a clear identification of the relevant anatomical points. The 45 X-rays were numerically anonymized. In this study, the authors agreed that a measurement difference between observers of ±5° was negligible and should not significantly impact the final results or, for example, the decision on surgical intervention. The angle consists of two lines. The distal line is drawn tangentially to the tibial plateau, extending from its cranial point to its most caudal point. The proximal line is drawn from the distal margin of the patellar articular surface to the caudal point of the distal line on the tibial plateau (Figure 2) [32].

Establishing the anatomical reference points on a dog stifle was a key factor when conducting this study. The tibial plateau was determined using the conventional Slocum method [33,34]. The tibial plateau was measured as a tangent to the medial tibial plateau comparable to the TPLO measurements (cranial to caudal point of plateau), strictly parallel to the tibial epiphyseal line [26,29]. We established the long axis of the tibia by connecting a line from the tibial intercondylar eminence to the center of the talar body.

The resulting TPPA was automatically calculated and documented as the TPPA measurement (Figure 3).

Three observers with varying levels of experience measured the TPPAs (using Portner and Pakzad’s method (2011) [29]) in the randomized X-rays in random order, twice and one month apart. Before commencing the measurements, all three observers underwent training on the anatomical points in various stifle X-rays.

One of the three observers was a young veterinarian with 4 years of experience in veterinary medicine (O1), another was an ECVS diplomate with over 30 years of surgical experience (O2), and the third had completed a surgical residency, with over 15 years of experience in imaging diagnostics (O3). The measurements were performed with RadiAnt™ DICOM Viewer (Version Nr: 5.0.2.21911, Copyright C 2009–2022 Medixant, Poznan, Poland) using the angle function.

### 2.2. Statistical Analysis

All the statistical analyses were performed using IBM SPSS v.27. Confirmation of a normal distribution was assessed using the Kolmogorov–Smirnov test. Besides descriptive statistics, we analyzed the agreement between the first and second TPPA measurements for every observer using the dependent-samples *t*-test. We used the interclass correlation coefficient (ICC) to determine the correlations between observers and analyze the agreement between the three observers regarding different stifle angles, weights, and the TPPA (Groups 1, 2, and 3). We used a mixed model to analyze the TPPA in the first and second measurement rounds for the different stifle angles and weight groups and to compare the differences within the stifle angle and weight groups. ANOVA was used to determine the correlations between the different stifle angles and weight groups. A *p*-value below 5% (*p* < 0.05) was considered significant for all the statistical analyses. A difference of ±5° between observers was not considered significant.

### 2.3. Results of the Cadaver Study

The tibial plateau–patella angle (TPPA) significantly differed between the three stifle angle groups (*p* < 0.001). The mean TPPA was 40.61 ± 4.21° in Group 1 (45°), 35.18 ± 4.21° in Group 2 (90°), and 27.92 ± 4.21° in Group 3 (135°). Figure 4 presents the TPPA measurement at various stifle angles. 

Group 1 exhibited significant differences compared with Group 2 (*p* = 0.002); Group 2 differed from Group 3 (*p* = 0.001); and there was a significant difference between Groups 1 and 3 (*p* = 0.001). The higher the stifle angle, the lower the TPPA. The inter-observer agreement was high (r = 0.852; *p* < 0.001). Moreover, the intra-observer agreement was high for the first (r = 0.931; *p* < 0.001), second (r = 0.876; *p* < 0.001), and third (r = 0.899; *p* < 0.001) observer. Observers 2 and 3 did not show a significant difference between the first and second measurements. The first observer showed a significant difference between their first and second measurements (r = 0.8769; *p* = 0.017), but it still fell within the tolerance range of ±5°. Nevertheless, the overall correlation for the first and second measurements across the three observers was not significant and showed a difference of SD ± 3.07°. The inter- and intra-observer variability was not statistically significant. The TPPA values decreased with a higher stifle angulation for all three observers (Figure 5).

The various stifle angle groups did not influence the measurements across all three observers. Furthermore, the weight group did not have an impact on the measurements by the observers. The measurements in the first and second rounds across the different stifle angles are presented in Figure 5. Body weight did not significantly (*p* = 0.347) influence the mean TPPA in any stifle angle group across all observers. The TPPA values across different weight groups and various stifle angles are presented in Figure 6 and Table 1. None of the groups showed significant variations in the mean TPPA values across the different stifle angle and weight groups. Specifically, the mean TPPA was 40.68° (SD ± 4.28°) in Group 1 (45 ± 5°); 35.26° (SD ± 3.88°) in Group 2 (90 ± 5°); and 28.23° (SD ± 3.54°) in Group 3 (135 ± 5°) (Table 1).

## 3. Study Part Two: Patient Study

### 3.1. Materials and Methods

In this part of the study, we analyzed archival radiographs based on the influence of the stifle angle on the TPPA in the cadaver study. We chose a positioning of 90 ± 5° because it is the most commonly used clinical projection for stifle joint diagnostics and has excellent inter- and intra-observer reliability. We applied the same inclusion and exclusion criteria as in the cadaver study. For instance, if a dog presented with hind leg lameness and we diagnosed cranial cruciate ligament rupture on one side, we utilized the radiograph of the contralateral (healthy) side for our measurements. We found the radiographs from previous examinations in our archive on optimal angulation and then checked our medical records to determine whether the patient had any current clinical stifle abnormalities. Additionally, a few radiographs from patients exhibited questionable tarsal or hip joint pathologies at the optimal stifle joint angle, allowing us to use both sides for our study. All these radiographs had to meet the same criteria as in the cadaver study to be included in the measurement process.

Therefore, 100 canine stifle radiographs with an angulation of 90 ± 5° were measured to evaluate the physiological limits of the TPPA. A total of 71 dogs were identified in the 100 X-rays, including 37 females (25 spayed) and 34 males (19 neutered). Of the 100 X-rays, 38 belonged to weight Group S, 24 to Group M, and 38 to Group L. The mean ± SD age of the dogs was 6.07 ± 3.5 years (1–13 years). The 100 X-rays from the archive were anonymized and measured in a randomized order once by the same three observers. As in the cadaver study, we allowed a difference of ±5° between observers because we believed this difference would not influence the results of surgical planning. We measured the radiographs under the same conditions and used the same weight categorization as in the cadaver study.

### 3.2. Statistical Analysis

We employed the ICC, Pearson’s correlation coefficient for the inter-observer agreement, and descriptive statistics to analyze the 100 radiographs. We used a mixed model to establish the relationships between observers and different weight groups. We used the same parameters as in the cadaver study. Moreover, we analyzed how many radiographic measurements exceeded the tolerance range of ±5° between observers within different weight groups for a more in-depth understanding.

### 3.3. Results of the Patient Study

The TPPA measurements in all 71 dogs (100 radiographs) at stifle angulation of 90 ± 5° ranged from 26.7° to 48.8° (Figure 7). The average TPPA for each observer (±SD) is presented in Table 2 and Figure 7.

The inter-observer agreement was high (r = 0.861; *p* < 0.001). The correlation between each pair of observers—Observers 1 and 2 (r = 0.858; *p* < 0.001), Observers 1 and 3 (r = 0.890; *p* < 0.001), and Observers 2 and 3 (r = 0.856; *p* < 0.001)—was also substantial.

This difference could also be observed when comparing how many of the 100 X-ray measurements between observers exceeded the tolerance range of ±5°. Just four (4%) radiographs of the 100 X-rays measured by Observers 1 and 2 exceeded the tolerance range of ±5° (one in Group L, two in Group M, and one in Group S). The difference was 14 out of 100 (14%) between Observers 1 and 3 (5 in Group L, 4 in Group M, and 5 in Group S). The largest difference of 20 out of 100 (20%) was observed between Observers 2 and 3 (6 in Group L, 4 in Group M, and 10 in Group S). Generally, the results did not significantly deviate from the tolerance range of ±5°. The variations in the differences between each observer and the mean deviations (MDs) are presented in Table 3.

## 4. Discussion

In this study, we aimed to investigate the influence of difference stifle angle groups on the TPPA, establish the physiological TPPA value range in dogs, and assess the inter- and intra-observer reliability of the measurements within various stifle angle and weight groups.

The TPPA values significantly changed at various stifle angles. Compared with studies in humans, different stifle angle groups significantly influenced the TPPA in dogs. This negative correlation (the lower the stifle angle, the higher the TPPA) was consistent and confirmed by all three observers. This can probably be explained by the greater distance between the tibial plateau and the patella at a lower stifle angle. We confirmed our hypothesis that the TPPA in dogs, unlike humans, depends on the stifle angulation.

Obtaining a high-quality radiograph without condylar overlay can be challenging in the 45° stifle position. It is often necessary to capture multiple radiographs to achieve an ideal image.

The patellar height position changes in relation to different tibial plateau angulations, and it has been suggested that a reduction in the tibial plateau increases the load on the patellar ligament. Consequently, the risk of patellar fractures increases as the tibia plateau angle decreases after TPLO [35].

In another study, different stifle angles (75°, 96°, 113°, 130°, and 148°) did not influence the measurements of the PLL-to-PL ratio [31], unlike our TPPA measurements. This represents a significant advantage of the PLL–PL ratio over TPPA measurements because angulation is not a limiting factor.

In a study on human stifles published in 2016, the TPPA was compared with other measurement methods such as the Insall–Salvati (I/S), Caton–Deschamps (C/D), and Blackburne–Peel (B/P) indices [32]. The results indicate that the TPPA values are consistent with those obtained using other methods but change if the surgery includes a tibial slope. Capkin and colleagues [36] reached the same conclusion when comparing the TPPA with the Insall–Salvati, modified Insall–Salvati, Blackburne–Peel, and Caton–Deschamps indices. The TPPA demonstrated higher reliability than the other methods.

The TPPA was pre- and post-operatively evaluated in dogs undergoing TPLO and tibial tuberosity advancement (TTA). The TPPA inconsistently correlated with the Insall–Salvati and Blackburne–Peel indices [26]. The main disadvantage of the study was the absence of a control group and a physiological TPPA value.

The TPPA in the dogs’ stifles showed a negative correlation with the Insall–Salvati index but a positive correlation with the differences in the Blackburne–Peel index. It is difficult to compare our results with those of this study because we lack information about the stifle angle in each case [37].

In another study, the proximodistal BPI, ISI, and CDI did not reliably distinguish between healthy stifle joints and joints with medial patellar luxation in small-breed dogs [38].

Small-breed dogs with second-grade medial patellar luxation had a more proximally positioned patellar ligament insertion compared to the control group. This is also a cause of the more proximally positioned patella in the femoral trochlea and probably plays an important role in the pathology of medial patellar luxation [39].

Unfortunately, there is no evidence of pathological TPPA values in dogs, so we can hardly direct compare our results with other studies on the proximodistal patella position.

The physiological TPPA values in our study ranged from 26.7° to 48.8°, with a mean value of 37.75° and MD of ±11.05°. We estimated the physiological TPPA range via our patient study in the 90 ± 10° stifle position. Based on these values, we postulate that angles below 26.7° could be classified as patella baja and angles above 48.8° as patella alta.

In her master’s thesis in 2019 [37], Winkler compared the pre- and post-operative (TPLO and TTA) TPPA, Insall–Salvati, and Blackburne–Peel indices in 39 different-sized dogs. She noted changes in the TPPA pre- and post-operatively, with the TPPA values ranging from 14.6° to 45.5° pre-operatively and from 14.6° to 51.4° post-operatively. This difference in the TPPA range may be explained by the cranial cruciate ligament or the low level of experience of one observer who performed the measurements. In Winkler’s study, the stifle angle ranged from 30° to 90° and did not influence the evaluation of the TPPA. The TPPA post-TPLO was higher in 35 dogs and lower in 4 dogs. This may have been due to the reduction in the TPA. The TPPA post-TTA was lower in 36 dogs and higher in 4 dogs. However, Winkler concluded that TPLO increases and TTA surgery decreases the TPPA value post-operatively.

Like our study on dogs, a study on the TPPA in cat stifles confirmed that an increasing stifle extension results in a reduction in the TPPA [40]. Our analysis indicates that the TPPA measurements were reproducible for all three observers, and they exhibited strong correlations and reproducibility in the cadaver and patient studies.

The stifle angle group did not influence the reproducibility of the TPPA measurements. As such, the observers could identify all three important anatomical points for measuring the TPPA at different stifle joint angles, but its value changed.

Only the third observer showed a significant difference of ±0.9° between the first and second measurements. However, considering our tolerance range of ±5°, this difference was not considered significant. This deviation in the third observer could be explained by them having less experience than Observers 1 and 2.

The key to obtaining correct and reproducible measurements was the exact identification of all three anatomical points: 1. The cranial edge of the tibial plateau, 2. the caudal tibial plateau, and 3. the distal end of the articular patellar surface. In our experience, the main difficulty is establishing the caudal point of the tibial plateau, where disagreement leads to differences in the TPPA measurements.

The second observer exhibited fewer differences in their measurements than the first and third observers. This could be attributed to the second observer’s higher experience level in establishing landmarks on the tibial plateau, as seen in tasks such as planning TPLO (tibial plateau leveling osteotomy) surgery. Determining the proximodistal patellar position can be complicated by factors such as osteophytes, an unusual patellar morphology, and an atypical distal attachment of the patellar ligament [14,20,22,30,41].

The examinations in the patient study revealed no significant differences between the observers, considering our tolerance range of ±5°. Observer 3 continuously measured higher angles than Observers 1 and 2. Observer 3 showed a larger deviation in the S group, which could be because anatomical deformities occur more often in small-breed dogs [3,42] or because the observer experienced more difficulty in identifying the caudal border of the tibial plateau in small-breed dogs. We consistently remained within our tolerance range, demonstrating the reproducibility of the TPPA measurements by the independent observers.

Evaluations of the TPPA in humans also revealed high inter- and intra-observer reliability [29,32,36].

In our experience, we can rely on measurements made by a single observer; however, additional training in small-breed dogs could be beneficial for precisely establishing anatomical points.

All the observers produced consistent TPPA measurements in the different weight groups (S, M, and L) without significant deviations; therefore, weight did not influence the identification of anatomical points. This is important for establishing future studies, for example, on breed-specific TPPA values. No difficulties in detecting the anatomical landmarks for the TPPA in humans have been reported, but poor-quality radiographs were also excluded from study [29].

Considering weight, the TPPA in the cadaver study did not change in relation to the S, M, or L groups. The small deviation in Group M could be due to the very small number of patients in this group (20%). The statistically visible variation in the S group could be explained by the larger deviation in the measurements performed by Observer 3.

Future studies should explore whether the TPPA is reliable for distinguishing between dogs with patellar luxation, cranial cruciate ligament rupture, and normal stifles. One possibility is to evaluate breed-specific TPPA values, particularly in small-breed dogs. A larger sample could allow us to analyze whether taking TPPA measurements in low-weight dogs are more challenging. These studies should compare the physiological reliability of the TPPA with that of other measurement methods.

## 5. Conclusions

Unlike in a human stifle, the TPPA varies at different stifle angles in dogs. The physiological range of the TPPA spans from 26.7° to 48.8°. The protocol for measuring the TPPA was proven to be reliable and reproducible for various observers. The observer’s level of experience may influence the achievement of precise TPPA measurements. Weight is not a significant factor for the successful measurement of the TPPA. Some observers may experience difficulty in establishing the caudal point of the tibial plateau in small-breed dogs.

## Figures and Tables

**Figure 1 animals-14-01798-f001:**
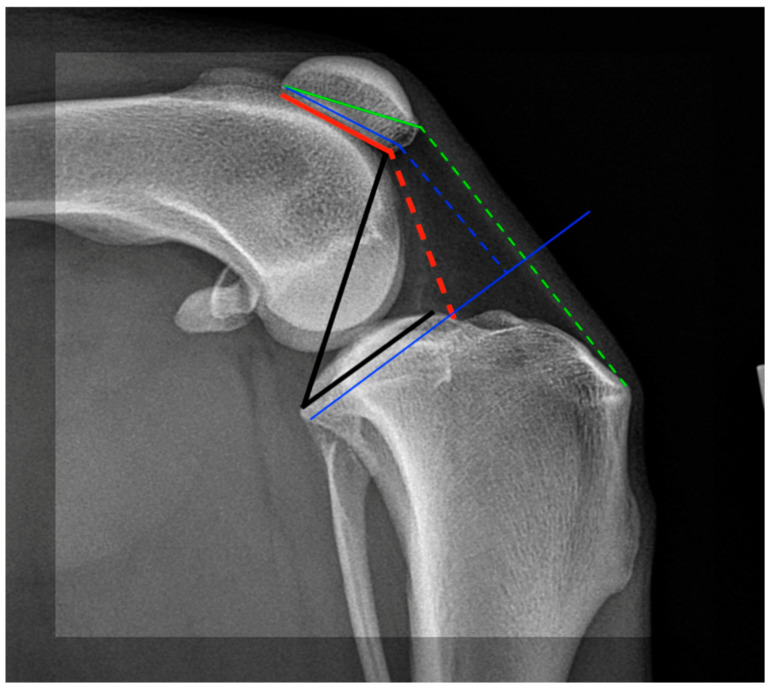
Simplified schematic presentation of the most commonly used indices for estimating patellar height on a dog’s stifle. Green lines represent the Insall–Salvati index (ISI: Solid line= patellar length; Dash line= patellar ligament lenght); red lines represent the de Carvalho index (CDI: Solid line= articular patellar length; Dash line= distance from distal articular surface to closest tibial cortex); blue lines represent the Blackburne–Peel index (BPI: proximal Solid line: articular patellar length; Dash line and distal Solid line: perpendicular distance to the distal articular surface from a line extended along the tibial plateau), black lines represent the tibial plateau–patella angle (TPPA: a detailed description follows in this study).

**Figure 2 animals-14-01798-f002:**
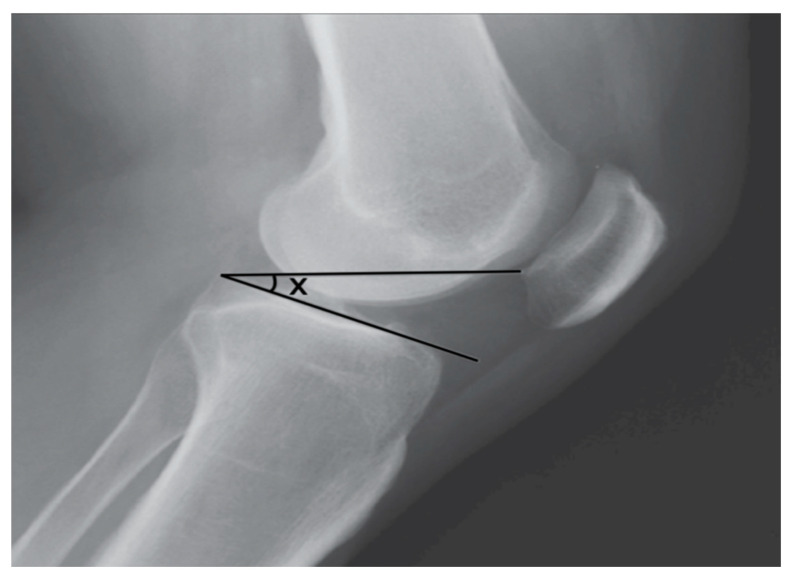
Two lines (distal and proximal) forming the TPPA in human knee.

**Figure 3 animals-14-01798-f003:**
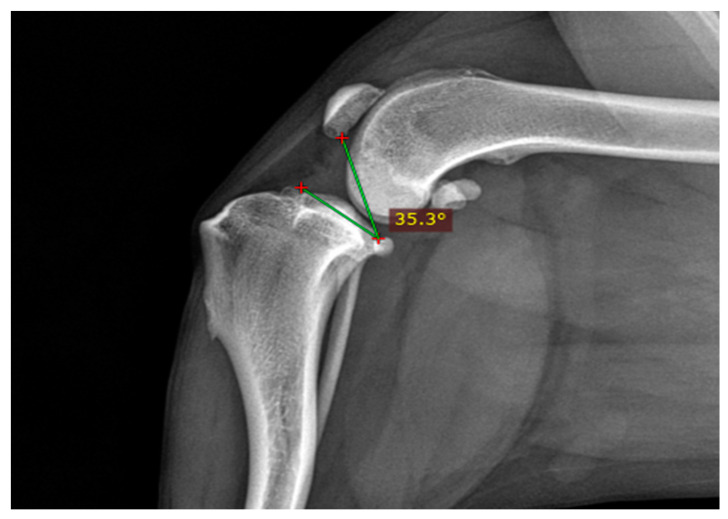
The distal line indicates the tibial plateau from its most cranial to its most caudal point. The proximal line extends from the caudal end of the tibial plateau to the distal end of the patellar articular surface.

**Figure 4 animals-14-01798-f004:**
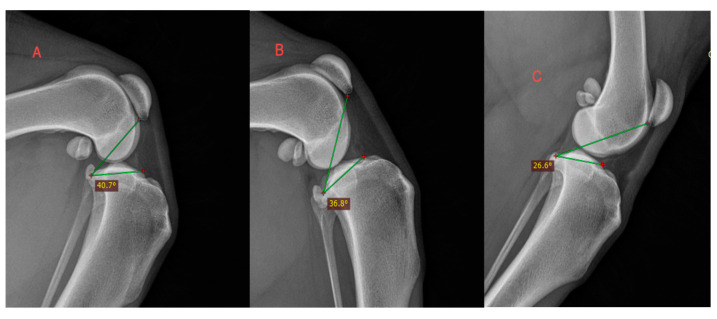
TPPA measurement at various stifle angles. (**A**) 45°; (**B**) 90°; (**C**) 135°.

**Figure 5 animals-14-01798-f005:**
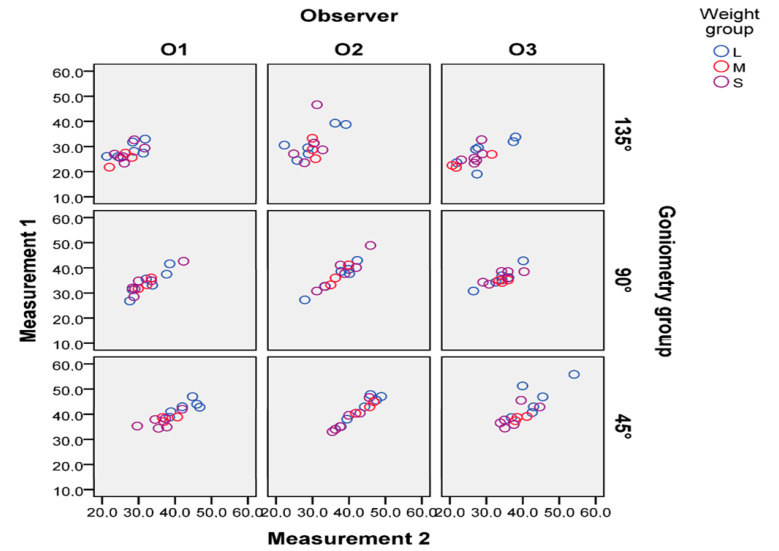
Measurements 1 and 2 across stifle angle and weight groups for all three observers. Values are presented in degrees on the vertical and diagonal axes. The reduction in TPPA values with higher stifle angulation is obvious and consistent across all three observers. Goniometry = stifle angle.

**Figure 6 animals-14-01798-f006:**
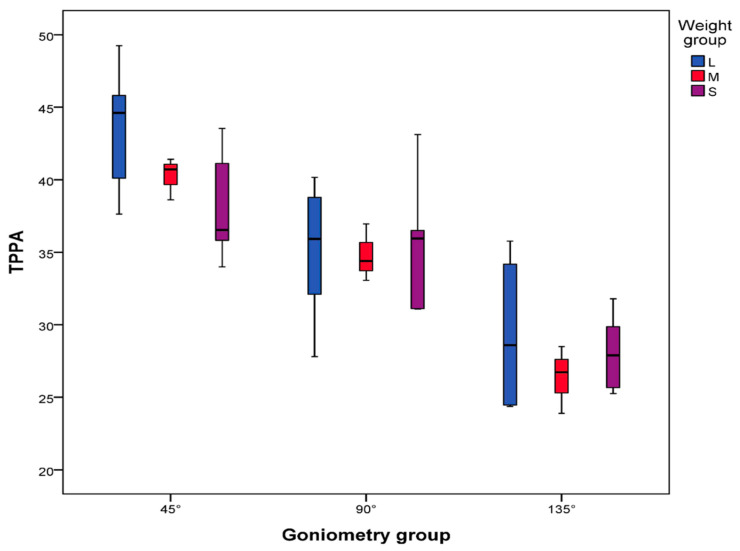
The TPPA in relation to the stifle and weight groups. The TPPA values are presented in degrees. Goniometry = stifle angle.

**Figure 7 animals-14-01798-f007:**
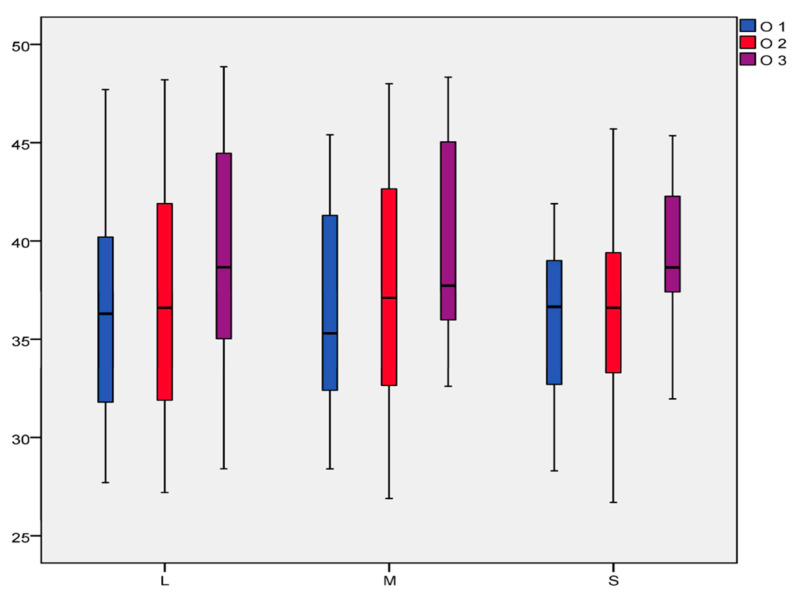
The range of TPPA values across three observers in different weight groups. TPPA values in degrees are presented on the vertical axis, and different weight groups are shown on the horizontal axis. Observer 1 (O1) is presented in blue, Observer 2 (O2) in red, and Observer 3 (O3) in purple.

**Table 1 animals-14-01798-t001:** Different weight groups across various stifle angles.

Weight Group	Stifle Angle	Mean TPPA	±SD ^1^
~45°	L	43.67°	
	M	40.28°	
	S	37.92°	
Across groups ^2^		**40.68°**	±4.28°
~90°	L	35.12°	
	M	34.80°	
	S	35.62°	
Across groups		**35.26°**	±3.88°
~135°	L	29.32°	
	M	26.37°	
	S	28.06°	
Across groups		**28.23°**	±3.54°

^1^ Standard deviation. ^2^ Mean TPPA across stifle angle groups.

**Table 2 animals-14-01798-t002:** Average TPPA with ±SD for each observer.

Observer	Mean TPPA	±SD
**1**	36.22°	±4.52°
**2**	37.10°	±5.29°
**3**	39.37°	±4.66°

**Table 3 animals-14-01798-t003:** Variations and mean deviations (MDs) between three observers.

Observer	Observer	Difference	MD
1	2	3.62–4.69°	4.15°
1	3	1.48–2.34°	1.95°
2	3	2.21–3.29°	2.75°

## Data Availability

The raw data supporting the conclusions of this article will be made available by the authors on request.

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
