# Peer review of "Evaluation of the Tibial Plateau–Patella Angle (TPPA) in Dogs"

_animals, 2024, doi:10.3390/ani14121798_

Round 1

Reviewer 1 Report

Comments and Suggestions for Authors

The title is confusing: Evaluation of Tibia-Plateau Angle (TPPA) in dogs

It is much better: Evaluation of Tibia-Plateau-Patella Angle (TPPA) in dogs

line 88: I think the sentence is not complete

line 91: A–P ratio, shoudl be  A-PL

Author Response

Dear Reviewer,

I am really grateful for your suggestions. I have made the necessary corrections, and I sincerely hope my revisions meet your expectations.

Sincerely,
Nedim Zaimovic

Reviewer 2 Report

Comments and Suggestions for Authors

Thank you for your manuscript. I have a few suggestions for areas to improve clarity.

Title: Please correct to read “tibial plateau-patella angle” 

I would suggest changing the word “goniometry” to stifle angles” or similar throughout the paper. The word goniometry refers to the act of measuring the stifle angle/range of motion which is not the context in which you have used it. 

Line 88: There is something missing from the beginning of this sentence. Please edit. 

Line 91: Please define L-P ratio and A-P ratio before using these abbreviations. 

Figure 1&2: I do not feel that it is necessary to include Figure 1 as the TPPA angle is comparably demonstrated in Figure 2 in the target species.  

For Figure 2, would it be possible to provide a more zoomed in image with the lines drawn and angle shown to improve clarity? 

Additionally for Figure 2, the notation of first line versus second line might be better described another way, such as the distal and proximal line. 

Line 204: Figure 2 is incorrectly referenced here, please correct. 

Line 232-233 and 240: Please edit to avoid unnecessary repetition discussing weight categorization/grouping. 

Line 280-289: These paragraphs would benefit from a bit further discussion as they seem disconnected from the rest of the discussion section.

Author Response

(The authors gave the same response as above.)

Reviewer 3 Report

Comments and Suggestions for Authors

Comments and Suggestions for Authors:

Line 2: The title is too simple and inaccurate. The first mistake is in the definition of the angle. According to the information provided in lines 29-30, the title should be "Evaluation of Tibial Plateau-Patella Angle (TPPA) in Dogs. "The authors describe a technique in dogs based on a human technique (Portner and Pakzad, 2011). A more attractive title would be: “Description of a new index for determining patellar height: Evaluation of Tibial Plateau-Patella Angle (TPPA) in Dogs."

Lines 29-30: In the abstract, the authors should specify the main objective of this new measure. Specifically, they should clarify whether the aim is to provide a more accurate assessment of patellar height compared to existing methods or to evaluate its potential clinical utility.

Lines 30-31: The authors state, "We conducted a two-step study on nine canine cadaver stifles without pathologies," but in reality, they only used 15 stifles.

Lines 41-42: The authors state, “Our protocol is reliable and reproducible, suggesting the potential benefits of training on the stifles of small-breed dogs”, but they should revise aspects of the methodology to confirm this claim.

Lines 43-44: Add the keyword "dog” and remove “inter- and intra-observer agreement”.

Lines 60-61: The authors state: “Patella alta is also associated with Osgood–Schlatter disease, a painful patellofemoral syndrome”, but it would have to be defined in the human species. Citation No. 15 is a work on cranial cruciate ligament rupture in dogs, it is the only work on animals. Does it comment anything on Osgood–Schlatter disease?

Lines 67-68: The authors state: “Patellofemoral OA is a more common cause of pain in the front of the stifle than OA in the tibiofemoral joint (19)”.If they are only referring to the human joint, the authors should clarify that it is in human medicine and should replace “stifle” with “knee”.

Lines 79-80: Replace “patellar ligament (PLL) length” for “patellar ligament length (PLL)”.

Lines 78-81: I believe it would be beneficial to incorporate a figure reflecting all these measurements (PLL, PL, A, PL), to understand the relationships between them. We would need to define the point on the femoral condyle for clarity.

Line 84: Replace “lateral X-ray stifle” for “mediolateral X-ray knee”.

Line 87: In the phrase “….in everyday practice. of TPPA measurements in dogs”, the point behind practice, what does it mean?

Line 91: What is L–P ratio and A–P ratio? Before using only abbreviations, the authors should define both ratios.

Lines 113-116: The authors should revise the order of the prompts: “Due to its easy applicability, this method may be useful in everyday practice, such as establishing a normal or pathological patellar height, in the pre-operative planning of patellar luxation surgery, or the insertion of a stifle endoprosthesis.

Lines 140-143: The text “Three observers with varying levels of experience measured the TPPAs (using Portner and Pakzad's method (2011)) in the randomized X-rays in a random order, twice, one month apart. Before commencing the measurements, all three observers underwent training on the anatomical points in various stifle X-rays” should be integrated with the last paragraph between lines 159-162.

Lines 153-158: The authors must define points A and B in this work to facilitate understanding of the procedure.

Line 163: Replace “Copyright C 2009-2022 Medixant” for “Copyright C 2009-2022 (Medixant, Poznan, Poland)”

Line 164: Figure number 2, which shows the mediolateral image of the stifle, must be enlarged so that only the stifle and the measurements made on it are visible. If the tibia is essential, it should be included, but the middle and proximal femur should be trimmed to focus solely on the knee and the measurements.

Lines 182-187. The authors should provide a mediolateral x-ray image of the same stifle, showing the TPPA obtained when the stifle is positioned at 45, 90, and 135 degrees.

Line 206: Replace “Fig. 4” for “Figure 4”.

Line 207: Replace “Tab. 1” for “Table 1”.

Line 249: The authors should make it clear that this range was obtained with a stifle angulation of 90º. Incorporate into the sentence.

Line 250: Replace “Tab. 2 and Fig. 5.” for “Table 2 and Figure 5”.

Line 268: Replace “Tab. 3” for “Table 3”.

Lines 274-275. The main objective should be to "establish the physiological TPPA value range in dogs," with the rest considered secondary objectives.

Lines 277-279. The authors should clarify the influence of the tibial plateau inclination with respect to its impact on the TPPA.

Lines 270-372 (Discussion): The authors do not compare their results with those obtained by applying other indices from different authors.

Line 281: What position is the 45º stifle position? The best projection to obtain an x-ray without overlapping of the femoral condyles is the mediolateral projection of the stifle at 135º of extension, achieved by moving the contralateral limb proximally.

Line 282: The ideal is to reduce radiation exposure to the animal, not on the staff. The studies should be performed under sedation to prevent owners or technicians from restraining the patient. The authors should confirm this fact and clarify it in the text.

Lines 286-289: The authors should clarify the origin of the following angles: 75°, 96°, 113°, and 130º.

Line 361: The sentence is incomplete.

Line 460: Replace “wtm 2022, 109” for “Wien Tierarztl Monat – Vet Med Austria.2022 ;109”

Author Response

(The authors gave the same response as above.)

Reviewer 4 Report

Comments and Suggestions for Authors

Dear Authors

Congratulations on your research. The paper is good. It would be good to expand/change a few points. Please consider these

1. line 154 -why is the word von here

2. It would be good to define a good lateral stifle image and have a comparison between good and bad?

3. How did the authors decide to define baja and alta as certain cut off angles?

4. It would be good to expand the potential application of these results and their importance to veterinary clinicians.

Best wishes

Reviewer

Author Response

(The authors gave the same response as above.)

Reviewer 5 Report

Comments and Suggestions for Authors

Thank you for the interesting study fassessing the distoproximal position of the patella using the TPPA.

I have the following remarks.

L2 Tibia-Plateau-Patella Angle. Patella is missing in the title.

L15 patellar height might be the correct terminilogy in bipeds but is suboptimal for quadrupeds. I would suggest distoproximal position in the femoral trochlea.

L18 Please write the term TPPA consistently throughout the manuscript

L19 You show that it is not independent of the stifle angle. Please correct

L22 Stifle angle not position. Please correct

L24 Without a description of what you consider to be the higher stifle angle this sentence is ambiguous. Clearer would be higher extension angle.

L25 the TPA is determined almost on a daily basis in veterinary orthopedic surgery in performing TBLO. Why should the caudal border suddenly caus problems? Please correct.

L31 are these opening or closing angles of the stifle joint? 90 degrees is clear but 45 and 135 ambiguous. Is 135 degrees conswidered the normal standing angle. Please clarify

L48 femoral trochlea instaed of groove

L52 CCL disease or deficiency  instead of rupture. Why introduce CCLD here. This does not relate to patellar position

L56 Reference 6 is not concerning praeop or postop assessment. Please correct

L61. Ref 15 is not concerning Osgood-Schlatter. Please correct. Schlatter is misspelled in the references

L66 infrapatellar fat pad. Please correct

L67 Ref 18 is missing Please correct

L76 dogs [22-24] Please correct [7, 21-24] in L75

L91 please explain the abbreviations L-P and A-P

L97 Please add Ref 34 here. No indication of patella alta in small breed dogs

L132 How many stifles per group? Please add

L148 Sequence of reference is incorrect 32-34 follow on 35

L156 which epiphyseal line are we considering here?

L185 You are comparing group 2+3 twice. Please correct

L220 Most used instead of usable?

L297-300 Is this relevant in view of your present study?

L307 ref 36 should be 38 I presume?

L334 cranial edge of the tibial plateau 

Please add and discuss the following reference on distoproximal position of the patella:

Liene Feldmane, Lars F H Theyse, Proximodistal and caudocranial position of the insertion of the patellar ligament on the tibial tuberosity and patellar ligament length of normal stifles and stifles with grade II medial patellar luxation in small-breed dogs. Vet Surg 2021 Jul;50(5):1017-1022. doi: 10.1111/vsu.13661. Epub 2021 May 20.

Comments on the Quality of English Language

Not applicable. Please see my remarks in the comments and suggestions for authors section

Author Response

(The authors gave the same response as above.)

Round 2

Reviewer 3 Report

Comments and Suggestions for Authors

Many of the suggested changes in the work have been made by the authors, but there are still some issues that need clarification before considering its acceptance.

“Comment 2: 2. Lines 29-30: In the abstract, the authors should specify the main objective of this new measure. Specifically, they should clarify whether the aim is to provide a more accurate assessment of patellar height compared to existing methods or to evaluate its potential clinical utility. 

Response 2: Thank you. I accept your suggestion and I made the necessary correction.

Old: We aimed to assess a new method, the tibial plateau–patella angle (TPPA).

New: Line 29-30: We aimed to evaluate the clinical applicability of a new method, the tibial plateau–patella angle (TPPA).”

Line 75: I think it would be interesting to include some clinical indications of the tibial plateau–patella angle (TPPA) in the abstract.

“Comment 3: Lines 30-31: The authors state, "We conducted a two-step study on nine canine cadaver stifles without pathologies," but in reality, they only used 15 stifles.

Response 3: Yes, we used 9 cadavers, however some of them had for example arthrosis on one of the stifles so we used just one without of any pathology. I hope the new version is better to understand:

Old: We conducted a two-step study on nine canine cadaver stifles without pathologies.

New: Line 30-32: We conducted a two-step study, first on nine canine cadavers (15 stifles) without pathologies, and, second using patient x-rays from archive.”

Lines 76-78: I suggest rewording the sentence to: “We conducted a two-step study, first on 15 stifles without pathologies from nine canine cadavers, and second, using 100 patient x-rays from the archive.”

“Comment 6: Lines 60-61: The authors state: “Patella alta is also associated with Osgood–Schlatter disease, a painful patellofemoral syndrome”, but it would have to be defined in the human species. Citation No. 15 is a work on cranial cruciate ligament rupture in dogs, it is the only work on animals. Does it comment anything on Osgood–Schlatter disease?

Response 6: I apologize for this mistake. An incorrect reference was included here, which was not related to Osgood-Schlatter disease. The reference has been removed.

Unfortunately, we did not find any relevant information correlating with Osgood-Schaltter disease in dog.”

Lines 171-172: I suggest rewording the sentence to: “Patella alta is also associated with Osgood–Schlatter disease, a painful patellofemoral syndrome in humans.”

“Comment 9: Lines 78-81: I believe it would be beneficial to incorporate a figure reflecting all these measurements (PLL, PL, A, PL), to understand the relationships between them. We would need to define the point on the femoral condyle for clarity.

Response 9: Thank you for the suggestion. We chose not to go into a detailed analysis (with figures) of other measurement methods. Instead, we plan to conduct a separate study to compare them to TPPA. The only important anatomical structure beside the tibia plateau is the distal articular border of the patella.”

Lines 187-192: My comment suggests including a figure to facilitate understanding of the different indices suggested by various authors. A comparative study is not requested; rather, the request is to include a figure that allows understanding of the different proposed indices. It would be similar to figure 1 of the work published by Miles et al (2012).

“Comment 18: Lines 182-187. The authors should provide a mediolateral x-ray image of the same stifle, showing the TPPA obtained when the stifle is positioned at 45, 90, and 135 degrees.

Response: Line 183-185 We found it unnecessary to place radiographs at different stifle angles because we proved that the identification of all relevant points is not influenced by varying stifle angles. All relevant anatomic points could be easily identified at any stifle angle by the three observers.”

Lines 446-447: My comment did not suggests identifying the landmarks in the three mediolateral x-ray images of the same stifle (45, 90, and 135 degrees). The anatomical points remain consistent. The aim was to show how the angles are modified by changing the angulation of the stifle.

“Comment 25: Lines 277-279. The authors should clarify the influence of the tibial plateau inclination with respect to its impact on the TPPA.

Response 25: I have made the necessary correction, and I hope this revised sentence provides a better explanation:

Old: This negative correlation was consistent and confirmed by all three observers. We confirmed our hypothesis that the TPPA in dogs, unlike humans, depends on the stifle angulation.

New: Line 284-288. This negative correlation (the lower the stifle angle, the higher the TPPA) was consistent and confirmed by all three observers. This can probably be explained by the greater distance between the tibial plateau and the patella at lower stifle angle.”

Lines 724-727: My comment suggested that the authors should clarify the influence of the tibial plateau angle (TPA) on the calculation of the tibial plateau–patella angle (TPPA). Currently, the calculation of the tibial plateau slope is crucial in techniques such as TPLO. In the angle defined by the authors, the first line refers to the tibial plateau without providing information about its inclination relative to the longitudinal axis of the tibial diaphysis. This could result in a significant calculation error of the angle studied by the authors. The authors should make it clear whether this aspect influences or does not influence the measured angle.

“Comment 28: Line 282: The ideal is to reduce radiation exposure to the animal, not on the staff. The studies should be performed under sedation to prevent owners or technicians from restraining the patient. The authors should confirm this fact and clarify it in the text.

Response 28: In our study, we only took x-rays on cadavers. I this case, radiation exposure was only relevant for researchers. We simply wanted to provide additional information based on our experienced while taking a radiograph of cadavers.”

Lines 775-776: I suggest rewording the sentence to: "It is often necessary to capture multiple radiographs to achieve an ideal image." I recommend removing the phrase "leading to longer radiation exposure for personnel". If they use cadavers, it is not necessary to restrain the animal, and if they are patients, they should be done under sedation.

New Commets:

Lines 759: Replace “knee” for “stifle”.

Lines 1025-200 (References): The references need improvement. The format in which they have been introduced is entirely different from the rest of the text.

Line 1195: Replace “… Patellar Height in Cats Usin the Tibia Plateau-Patella Angle….” for “…..Patellar Height in Cats Using the Tibia Plateau-Patella Angle….”

Author Response

Dear Reviewer,

Thank you for investing your time in optimizing my manuscript.

Sincerely, Nedim Zaimovic

Reviewer 5 Report

Comments and Suggestions for Authors

Thank you for the thorough revision

Comments on the Quality of English Language

Not applicable

Author Response

(The authors gave the same response as above.)
